# Extinction Risk Assessment and Chemical Composition of Aerial Parts Essential Oils from Two Endangered Endemic Malagasy *Salvia* Species

**DOI:** 10.3390/plants12101967

**Published:** 2023-05-12

**Authors:** Stéphan R. Rakotonandrasana, Mathieu Paoli, Mamy J. Randrianirina, Harilala Ihandriharison, Marc Gibernau, Ange Bighelli, Marrino F. Rakotoarisoa, Pierre Tomi, Charles Andrianjara, Félix Tomi, Delphin J. R. Rabehaja

**Affiliations:** 1Department of Ethnobotany and Botany, National Center for Applied Pharmaceutical Research, Antananarivo 101, Madagascar; stephanandrasana@gmail.com (S.R.R.); rino2r@hotmail.fr (M.F.R.); 2Laboratoire Sciences Pour l’Environnement, Université de Corse-CNRS, UMR 6134 SPE, Route des Sanguinaires, 20000 Ajaccio, France; paoli_m@univ-corse.fr (M.P.); gibernau_m@univ-corse.fr (M.G.); bighelli_a@univ-corse.fr (A.B.); tomi_p@univ-corse.fr (P.T.); 3Département Phytochimie et Contrôle Qualité, Institut Malgache de Recherches Appliquées (IMRA), Antananarivo 102, Madagascariharilalandr@gmail.com (H.I.); charles.andrianjara@gmail.com (C.A.); rabehaja@yahoo.fr (D.J.R.R.)

**Keywords:** *Salvia sessilifolia*, *Salvia leucodermis*, essential oil, threat, Madagascar

## Abstract

Seven essential oil samples of two endemic species of Malagasy sage, *Salvia sessilifolia* Baker and *Salvia leucodermis* Baker, were investigated via GC(RI), GC-MS and ^13^C NMR spectrometry. In total, 81compounds were identified accounting for 93.5% to 98.7% of the total composition. The main constituents for the both species were (*E*)-β-caryophyllene (29.2% to 60.1%), myrcene (1.2% to 21.7%), α-humulene (5.2% to 19.7%), (*E*)-nerolidol (0.8% to 15.5%) and caryophyllene oxide (1.4% to 10.8%). Ethnobotanical survey of 46 informants revealed that decoctions of leafy twigs and chewed leaves were usually used. Due to the repeated fires, over-harvesting and grazing, the populations of *S. sessilifolia* and *S. leucodermis* are drastically fragmented. These risk factors led to threats to the habitats of the target species. *Salvia sessilifolia* Baker and *Salvia leucodermis* Baker are proposed to be classified as endangered species.

## 1. Introduction

One of the largest plant families in the world, Lamiaceae contains 7173 species belonging to 236 genera [1]. *Salvia* (Sage) is one of these genera, and comprises 980 species distributed in tropical and temperate regions. Recent phylogenetical studies have split this genus into four clades and five additional genera. However, Malagasy species remain within the *Salvia* genus and represent six endemic species [2].

In the phylogenetic study at the level of the genus *Salvia* worldwide [2], the endemic species *S. sessilifolia* and *S. leucodermis* are closely related to species from East Africa. Indeed, subclade I-A contains two species from East Africa (*S. nilotica* and *S. somalensis*) and all Madagascar and South African species. The clade is poorly resolved with only a few small, supported terminal clades, of which one comprises *S. sessilifolia* and *S. leucodermis.*


According to Hedge et al. (1998), *Salvia sessilifolia* Baker and *Salvia leucodermis* Baker consist of sub-shrubs up to 1.50 m high, with erect, woody stems [3]. The leaves and young twigs are aromatics. The leaves are simple, opposite and without stipules. The fruit bears an accrescent and spreading-deflected calyx.

Concerning *Salvia sessilifolia* Baker, the leaves are sessile, rough, ascending–erect, linear–oblong to oblanceolate, cuneate at the base, crenate and slightly revolute on the edge. The upper sides of the leaf blade are green with dense, stiff, shiny hairs; the undersides are very densely haired, white and not glandular. The inflorescences are short, carry 4 to 10 flowers, are more or less close together, and are usually purple or reddish-purple (Figure 1).

*Salvia leucodermis* Baker possesses dense, close together, silvery-white, oblong–elliptical to obovate leaves, which are velvety on both sides but very dense on the underside. The petioles do not exceed 7 mm. The inflorescences bring 8 to 10 clustered flowers, which are usually white, and rarely purplish pink. (Figure 1).

Madagascar has 3245 species of medicinal plants, 60% of which are endemic. Surveys on the use of medicinal plants have been carried out; the family of Lamiaceae is among the top 30 most cited, and its frequency of citation in relation to the surveys is 52.9% [4].

In recent decades, the conservation of endemic and threatened species has received more attention throughout the world. Studies on the genetic diversity of endangered species have increased in recent years due to their central importance in planning both in situ and ex situ conservation efforts [5]. However, chemical diversity is also important to evaluate the influence of risk factors and the study of chemical composition of essentials oil should be a complementary approach. To our knowledge, no data are reported concerning these two sage endemic species.

Ecological factors refer to habitat elements that directly or indirectly affect the growth, development, reproduction, behavior and distribution of plants in the environment. For Madagascar, the global change mainly is associated with biological factors: animal factor (grazing), anthropization (fire, over-harvesting, forest degradation).

Our aim is to report, for the first time, the empirical uses and the essential oil components of two sage Malagasy endemics species, *Salvia sessilifolia* and *Salvia leucodermis*, and to evaluate their status of conservation. On the one hand, we have studied the population status of these two species in relation to their ecology. On the other hand, we took the opportunity to study the chemical composition of its EO with limited impact on the populations. In the situation that an original/interesting EO composition was found, it would have been a supplementary argument for its preservation. Our study was carried out with reference to the following research permissions delivered by the ministry of environment and sustainable development: n° 38/14/MEEF/SG/DGF/DCB/SAP/SCB; n° 28/16/ MEEMF/SG/DGF/DAPT/SCB.Re; n° 38/17/ MEF/SG/DGF/DSAP /SCB.Re; N° 160/22/ MEDD/SG/DGGF/DAPRNE/ SCBE.Re.

## 2. Results and Discussion

### 2.1. Ethnobotanical Survey

Both species are known as Tsiparapanda or Tsiparapandy. Sometimes, *Salvia sessilifolia* Baker is called Tsiparapandalahy or Tsiparapandamena, while *Salvia leucodermis* Baker is called Tsiparapandafotsy.

Ethnobotanical survey of 46 informants revealed that the uses of these two species remain the same (Figure 2). The decoction of leafy twigs has oxytocic and antitussive properties and helps prevent certain diseases or relieve fatigue. Chewed leaves are applied to wounds, abscesses or insect bites. Its leafy branches are also sold at medicinal plant merchants around the Ankaratra massif (ANK), namely, in Antsirabe, Ambatolampy, Faratsiho (FAR), Andranomiely and Manalalondo (ITS), and generated an over-harvesting.

### 2.2. Ecology and Risk of Extinction

*S. sessilifolia* Baker is mainly distributed around the Ankaratra (ANK) and Ibity (IBT) massifs, above 1300 m altitude, on rocky slopes, on wooded grassland-bushland mosaic and in *Uapaca bojeri* Bail. forest while *S. leucodermis* Baker is found in the wooded grassland-bushland mosaic of the Ankaratra massif and those of Andringitra (ANG), above 1300 m altitude (Figure 2).

Based on 25 herbarium specimens, the results of the analysis of the parameters according to the assessment of the extinction risk of these two species are summarized in Table 1.

Subpopulations outside and inside of the protected areas are subject to repeated fires. In the protected Manjakatompo–Ankaratra (M-ANK) area and that of Andringitra (ANG), fires are accentuated by grazing. The subpopulations outside the protected areas are threatened by the extension of agricultural lands and also grazing.

Consequently, the subpopulations of the two-target species are severely fragmented; both extent of occurrence was estimated to be less than 100 km^2^ with most (≥65%) subpopulations found outside protected areas (Table 1). Moreover, continuous decline was observed for both species in the area of occupancy and occurrence, but also in the habitat.

### 2.3. Chemical Composition of Essential Oils

The yields calculated (Table 2) from fresh material (*w*/*w*) easily discriminated the two species: 0.11–0.23% (*S. sessilifolia)* vs. 0.31–0.41% (*S. leucodermis)*. 

The chemical composition of the seven oil samples collected in four locations was dominated by sesquiterpene hydrocarbons. Among them, (E)-β-caryophyllene (29.2–60.1%) was the major compound (Table 3 and Table 4). The chromatogram of the Sle1 sample (non-polar column) along with the main components is presented in Figure 3.

Other compounds were present in appreciable amounts: α-humulene (5.2–19.7%), (*E*)-nerolidol (0.8–15.5%) and caryophyllene oxide (3.6–10.8%). Among the monoterpenes, myrcene ranged from 1.2% to 21.7%. The overlapped peaks of cascarilladiene (synonym: eudesma-5,7-diene) and (*E*)-β-caryophyllene on non-polar column confirmed that the use of two columns of different polarity for the analysis of an essential oil is mandatory. Herein, the identification of cascarilladiene was achieved using ^13^C NMR reference spectra compiled in the laboratory spectral library.

The chemical composition of the seven oil samples was homogenous and not related to the (i) species, (ii) altitude or (iii) location. All the determined chemical compositions were dominated by (*E*)-β-caryophyllene (29.2–60.1%). However, Sse3 and Sle3 samples were slightly different and exhibited high amounts of (*E*)-nerolidol (15.5%) and myrcene (21.7%), respectively. It is noticeable that acora-3,10(14)-diene and ε-cadinene were identified only in the Sse1 sample with amounts close to 3%.

(*E*)-β-caryophyllene, was reported as a main component for several *Salvia* species from Colombia, Iran, Turkey and Spain [14]. However, the chemical compositions of *S. sessilifolia* and *S. leucodermis* were drastically different from those of East African species [15,16,17,18,19,20] (Appendix A). Among the sixteen chemical compositions reported in the literature for 15 species; the percentage of (*E*)-β-caryophyllene ranged from 0 to 13.1%, and the major components were the mainly oxygenated compound: 1,8-cineole (40.5%; *S. chamelaegnea*) [19], linalool (44.4%; *S. schimperi*) [16], geraniol (19.6%; *S. dolomitica* in South Africa), linalyl acetate (19.6%; *S. dolomitica* in South Africa) [20], bornyl acetate (16.2%; *S. somaliensis*) [17], caryophyllene oxide (22.6%; *S. radula*) [19], spathulenol (29.1%; *S. africana-caerulea*) [19], viridiflorol (24.5%; *S. albicaulis*) [20], and α-bisabolol (65.5%; *S. runcinata*) [20].

## 3. Materials and Methods

### 3.1. Plant Material

Samples were collected outside of the protected areas of Manjakatompo-Ankaratra and Andringitra. The data related to the sampling were reported in Figure 2 and Table 4. Vouchers were deposited in the herbarium of medicinal plants of Madagascar (CNARP). Botanical identification was carried out by the first author based at the CNARP herbarium.

### 3.2. Extinction Risk Assessment

The extinction risk assessment is based on the UICN Red List criteria for plant species, version 3.1 [21]. Geographic range (criteria B) was used to estimate the extinction risk. Geographical localities of previous collections of the two species were consulted in the (i) herbarium of medicinal plants of Madagascar (CNARP), (ii) online herbarium of the MNHN, (iii) herbaria of the Botanical and Zoological Park of Tsimbazaza and (iv) Tropicos database site.

The number of the sub population (geographically or otherwise distinct groups in the total number of individuals of the taxon between which demographic or genetic exchange are very little), extent of occurrence (the area limited within the shortest continuous imaginary boundary which can be drawn to encompass all the known, inferred or projected sites of present occurrence of a taxon, excluding cases of vagrancy) and area of occupancy (the area occupied by a taxon within its extent of occurrence, excluding cases of vagrancy) were determined by using Arc view software. Threat or Declin population (DP) were calculated according to the formula:$$DP\left(\%\right)=\frac{Numberofsubpopulationsoutsideoftheprotectedarea}{Totalnumberofthesubpopulations}\times 100$$

Botanical surveys were carried out in the massif of Ankaratra and its surroundings and in that of Andringitra to obtain the data on distribution, in order to know the states of the habitats, and to determine the threats to the species and its uses. Voucher specimens were made during the fieldwork and deposited in the herbarium of medicinal plants of Madagascar.

During the botanical surveys, the information on traditional uses were collected from 46 persons: 26 around the Ankaratra massif and 20 on the northeastern slope of the Andringitra massif.

### 3.3. Essential Oil Isolation

For each species, individual samples (aerial parts) were collected in a limited area, early in the morning and in dry weather. Aerial parts (200 g) were submitted to hydrodistillation for 2 h 30 with a Clevenger-type apparatus in a 1 L flask. The yields were calculated from fresh material (*w/w*) (Table 2). The essential oil samples obtained were conserved at 4 °C.

### 3.4. Gas Chromatography and Gas Chromatography–Mass Spectrometry in Electron Impact Mode

GC analyses were performed on a Clarus 500 PerkinElmer Chromatograph (PerkinElmer, Courtaboeuf, France), equipped with a flame ionization detector (FID) and two fused-silica capillary columns (length, 50 m; diameter, 0.22 mm; film thickness, 0.25 µm), BP-1 (polydimethylsiloxane) and BP-20 (polyethylene glycol). The oven temperature was programmed from 60 °C to 220 °C at 2 °C/min and then held isothermal at 220 °C for 20 min; injector temperature: 250 °C; detector temperature: 250 °C; carrier gas: hydrogen (0.8 mL/min); split: 1/60; injected volume: 0.5 µL. The relative proportions of the oil constituents were expressed as percentages obtained by peak-area normalization, without using correcting factors. Retention indices (RI) were calculated relative to the retention times of a series of *n*-alkanes (C8–C29) with linear interpolation («Target Compounds» software from PerkinElmer).

GC-MS analyses were performed on a Clarus SQ8S PerkinElmer TurboMass detector (quadrupole), directly coupled with a Clarus 580 PerkinElmer Autosystem XL (PerkinElmer, Courtaboeuf, France), equipped with a BP-1 (polydimethylsiloxane) fused-silica capillary column (lenght, 50 m; diameter, 0.22 mm; film thickness 0.25 µm). The oven temperature was programmed from 60 to 220 °C at 2°/min and then held isothermally for 20 min; injector temperature, 250 °C; ion-source temperature, 250 °C; carrier gas, Helium (1 mL/min); split ratio, 1:80; injection volume, 0.5 µL; ionization energy, 70 eV. The electron ionization (EI) mass spectra were acquired over the mass range 35–350 Da.

### 3.5. Nuclear Magnetic Resonance

All nuclear magnetic resonance (NMR) spectra were recorded on a Bruker AVANCE 400 Fourier Transform spectrometer (Bruker, Wissembourg, France) operating at 100.623 MHz for ^13^C, equipped with a 5 mm probe. The solvent used was CDCl_3_, with all shifts referred to internal tetramethyl silane (TMS). ^13^C NMR spectra of the oil samples were recorded with the following parameters: pulse width, 4 µs (flip angle 45°); relaxation delay D1, 0.1 s; acquisition time, 2.7 s for a 128 K data table with a spectral width of 25,000 Hz (250 ppm); CPD mode decoupling; digital resolution, 0.183 Hz/pt. The number of accumulated scans was 3000 for each sample (30 mg in 0.5 mL of CDCl_3_).

### 3.6. Identification of Individual Components

Identification of the individual components was carried out as follows: (i) by comparison of their GC retention indices on non-polar and polar columns, with those of reference compounds [6,7,10]; (ii) on computer matching against commercial mass spectral libraries [7,22,23]; (iii) on comparison of the signals in the ^13^C NMR spectra of the samples with those of reference spectra compiled in the laboratory spectral library, with the help of laboratory-made software [24,25,26]. This method allowed the identification of individual components of the essential oil at contents as low as 0.4%.

## 4. Conclusions

The essential oils of aerial parts of *Salvia sessilifolia* Baker and *Salvia leucodermis* Baker were characterized by the presence, of (*E*)-β-caryophyllene, which, as a major component, possesses several important pharmacological activities, ranging from pain treatment to neurological and metabolic disorders.

According to the results of the parameters assessed and the observed threats on the habitats and our botanical prospections, the target species, *Salvia sessilifolia* Baker and *Salvia leucodermis* Baker are proposed to be classified as endangered species or EN *B2ab(ii, iii, iv).*

## Figures and Tables

**Figure 1 plants-12-01967-f001:**
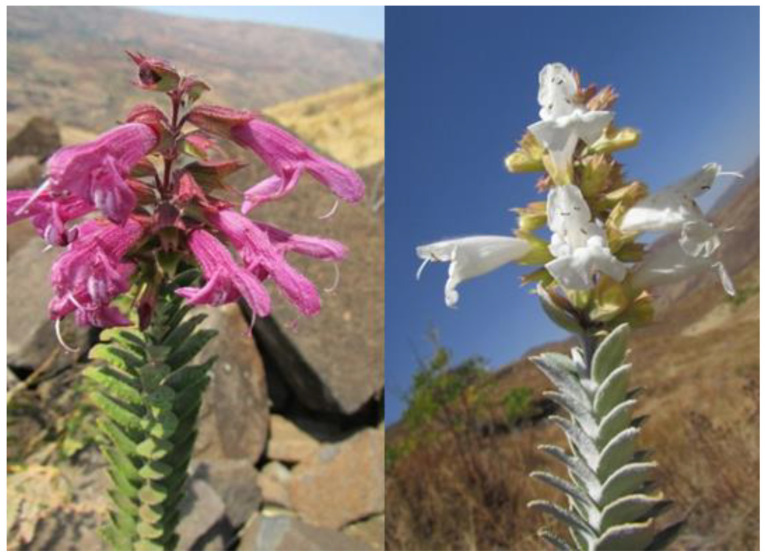
*Salvia sessilifolia* (**left**) and *Salvia leucodermis* (**right**).

**Figure 2 plants-12-01967-f002:**
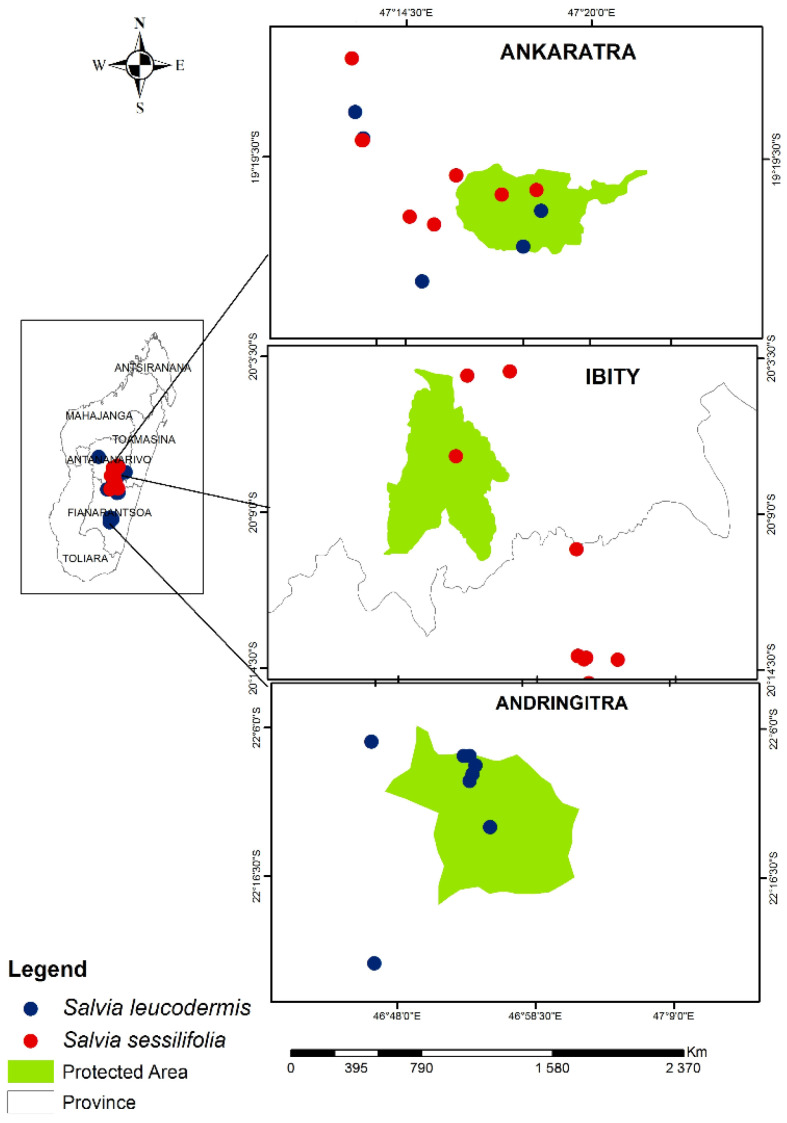
Collection maps of the two Salvia species.

**Figure 3 plants-12-01967-f003:**
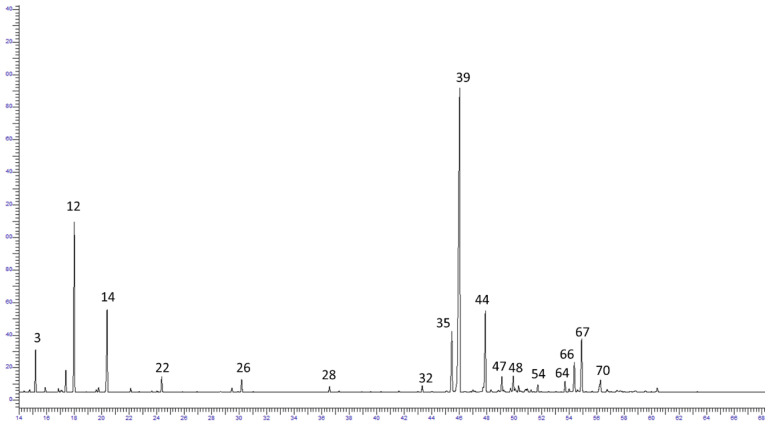
Chromatogram of Sle1 sample (BP-1, non-polar column).

**Table 1 plants-12-01967-t001:** Parameters for assessing the conservation status.

	*S. sessilifolia*	*S. leucodermis*
Number of subpopulations	14	17
Subpopulation inside of Protected Area	3 (M-ANK, IBT)	6 (M-ANK, ANG)
Extent of occurrence (km^2^)	5539	31,272
Area of occupancy (km^2^)	72	84
Declin population (Threat prediction %)	78.6	64.7
Number of Herbarium specimens	25	25

**Table 2 plants-12-01967-t002:** Essential oil yield.

Sample	Aerial Parts % (*w*/*w*)						
	Sse1	Sse2	Sse3	Sse4	Sle1	Sle2	Sle3
*S. sessilifolia*	0.14	0.23	0.11	0.16			
*S. leucodermis*					0.41	0.36	0.31

**Table 3 plants-12-01967-t003:** Chemical composition of oil samples of *S. sessilifolia* and *S. leucodermis*.

N°	Components ^a^	Ria ^b^	Rip ^b^	^c^ RI_lit_	Sse1 ^k^	Sse2	Sse3	Sse4	Sle1 ^l^	Sle2	Sle3	Identification ^m^
1	(Z)-Hex-3-en-1-ol	832	1377	842	-	tr	tr	tr	-	tr	tr	RI, MS
2	α-Thujene	922	1016	926	-	tr	tr	-	0.2	tr	0.1	RI, MS
3	α-Pinene	932	1016	934	0.1	0.6	0.1	0.6	3.3	0.6	0.4	RI, MS, ^13^C NMR
4	Camphene	942	1066	947	-	tr	-	-	0.4	0.1	-	RI, MS, ^13^C NMR
5	Oct-1-en-3-ol	958	1439	965	-	0.1	0.1	tr	0.2	0.2	0.4	RI, MS, ^13^C NMR
6	Octan-3-one	961	1255	966	-	0.1	-	-	0.1	tr	0.1	RI, MS
7	Sabinene	963	1122	967	0.1	tr	0.1	tr	0.2	-	tr	RI, MS
8	β-Pinene	968	1112	973	tr	0.1	0.1	tr	1.7	0.2	0.2	RI, MS, ^13^C NMR
9	Myrcene	979	1162	983	1.2	11.9	7.7	2.2	12.6	15.0	21.7	RI, MS, ^13^C NMR
10	α-Phellandrene	995	1164	999	0.3	tr	0.1	tr	tr	tr	tr	RI, MS
11	δ-3-Carene	1005	1149	1007	tr	tr	tr	tr	tr	tr	0.1	RI, MS
12	α-Terpinene	1007	1181	1011	0.1	0.1	tr	tr	0.2	0.1	0.1	RI, MS
13	*p*-Cymene	1010	1271	1015	0.1	0.2	0.1	0.1	0.3	0.2	0.2	RI, MS
14	Limonene *	1020	1202	1023	0.2	8.4	1.4	0.4	6.2	2.4	tr	RI, MS, ^13^C NMR
15	β-Phellandrene *	1020	1206	1021	-	-	-	-	-	-	1.5	RI, MS, ^13^C NMR
16	1,8-Cineole *	1020	1211	1022	0.4	0.2	3.7	0.9	0.8	0.2	0.3	RI, MS, ^13^C NMR
17	γ-Terpinene	1046	1244	1050	0.1	0.1	0.1	0.1	0.3	0.1	0.1	RI, MS
18	*trans*-Sabinene hydrate	1052	1470	1056	-	-	-	-	tr	-	-	RI, MS
19	*cis*-Linalool oxide THF	1055	1442	1065	-	tr	tr	tr	0.1	0.2	0.1	RI, MS
20	*trans*-Linalool oxide THF	1072	1470	1072	-	-	-	-	-	0.2	0.1	RI, MS
21	Terpinolene	1078	1282	1079	0.1	tr	0.2	0.2	tr	0.1	0.1	RI, MS
22	Linalool	1082	1546	1086	0.3	1.8	1.7	1.3	1.1	2.3	2.8	RI, MS, ^13^C NMR
23	Borneol	1148	1684	1153	-	-	-	-	0.1	-	-	RI, MS
24	Terpinen-4-ol	1159	1602	1164	0.5	0.1	0.1	0.2	0.3	0.3	0.2	RI, MS, ^13^C NMR
25	Methyl salicylate	1166	1780	1170	-	0.1	-	-	0.1	0.1	0.1	RI, MS
26	α-Terpineol	1170	1690	1176	0.3	0.9	0.5	0.9	0.9	6.4	1.6	RI, MS, ^13^C NMR
27	Carvone	1215	1737	1218	-	-	-	-	tr	tr	-	RI, MS
28	Bornyl acetate	1268	1578	1270	-	-	-	0.4	0.4	-	-	RI, MS, ^13^C NMR
29	Myrtenyl acetate	1303	1676	1305	-	-	-	-	tr	tr		RI, MS
30	α-Cubebene	1346	1455	1352	0.1	-	-	0.7	0.1	-	tr	RI, MS, ^13^C NMR
31	α-Ylangene	1369	1471	1370	0.1	-	-	0.2	0.1	-	0.1	RI, MS
32	α-Copaene	1373	1488	1375	0.4	0.1	0.8	2.5	0.5	0.1	0.1	RI, MS, ^13^C NMR
33	α-Bourbonene	1381	1513	1378 ^d^	tr	-	0.1	-	tr	-	-	RI, MS
34	Isocaryophyllene	1401	1870	1405 ^e,f^	tr	0.2	-	0.3	0.1	-	-	RI, MS
35	α-Gurjunene	1407	1524	1405	0.1	tr	0.1	0.3	5.7	0.3	0.7	RI, MS, ^13^C NMR
36	Aristolene	1413	1567	1416 ^f^	-	-	1.0	-	-	-	-	RI, MS, ^13^C NMR
37	(*Z*)-β-Farnesene	1417	1633	1418 ^d^	-	-	-	-	-	-	0.6	RI, MS, ^13^C NMR
38	Cascarilladiene *	1418	1753	1416 ^d^	3.4	0.1	1.0	1.5	0.8	0.1	0.8	RI, MS, ^13^C NMR
39	(*E*)-β-Caryophyllene *	1418	1599	1419	47.6	60.1	29.2	43.8	37.3	36.2	43.3	RI, MS, ^13^C NMR
40	β-Copaene	1423	1583	1430 ^d^	0.1	0.2	-	-	tr	-	-	RI, MS
41	(*E*)-*α*-Bergamotene	1430	1586	1434 ^d^	0.3	tr	tr	0.1	0.1	tr	tr	RI, MS
42	α-Guaiene	1432	1590	1442	0.3	-	-	-	0.2	0.2	0.2	RI, MS
43	(*E*)-β-Farnesene	1446	1665	1449	0.8	0.2	tr	0.6	-	0.3	0.2	RI, MS, ^13^C NMR
44	α-Humulene	1448	1665	1449	19.7	5.5	10.8	13.8	7.0	5.2	5.5	RI, MS, ^13^C NMR
45	Acora-3,10(14)-diene	1453	1650	1457 ^d^	2.1	-	-	-	-	-	-	RI, MS, ^13^C NMR
46	γ-Muurolene	1467	1688	1473	-	-	-	-	-	-	0.1	RI, MS
47	α-Curcumene	1468	1769	1473 ^d^	0.3	0.1	1.1	3.0	1.2	1.0	0.1	RI, MS, ^13^C NMR
48	γ-Himachalene	1471	1688	1471	-	-	-	-	-	0.8	0.3	RI, MS, ^13^C NMR
49	Germacrene D	1475	1709	1475	-	-	-	0.2	-	-	-	RI, MS
50	β-Selinene	1479	1713	1480	0.4	-	tr	0.3	0.3	tr	-	RI, MS, ^13^C NMR
51	ε-Cadinene	1481	1690	1483 ^d^	2.9	-	-	-	-	-	-	RI, MS, ^13^C NMR
52	α-Zingiberene	1485	1716	1482	-	-	1.1	2.3	-	0.3	1.0	RI, MS, ^13^C NMR
53	Ledene	1489	1682	1491 ^d^	-	-	0.2	0.4	-	-	-	RI, MS, ^13^C NMR
54	α-Selinene	1491	1715	1494 ^d^	0.1				1.0	0.1	1.2	RI, MS, ^13^C NMR
55	α-Muurolene	1496	1718	1496 ^d^	0.3	-	0.2	0.5	-	-	-	RI, MS, ^13^C NMR
56	α-Bulnesene	1496	1711	1500	0.6	-	-	-	-	-	-	RI, MS, ^13^C NMR
57	β-Bisabolene *	1498	1722	1500	-	0.1	0.1	0.4	0.3	1.1	0.1	RI, MS, ^13^C NMR
58	β-Curcumene *	1498	1738	1503	-	-	-	-	-	0.9	-	RI, MS, ^13^C NMR
59	γ-Cadinene	1503	1752	1505	0.2	tr	0.4	1.0	0.2	tr	-	RI, MS, ^13^C NMR
60	Calamenene #	1507	1827	1509	tr	-	-	0.2	tr	0.1	0.1	RI, MS
61	δ-Cadinene	1513	1752	1513	0.6	0.1	0.9	3.0	0.6	-	-	RI, MS, ^13^C NMR
62	β-Sesquiphellandrene	1514	1762	1513	-	-	-	-	-	0.6	-	RI, MS, ^13^C NMR
64	(*E*)-Nerolidol	1546	2039	1550	2.6	4.9	15.5	3.1	0.8	8.9	5.0	RI, MS, ^13^C NMR
65	Caryolan-8-ol	1556	2046	1567 ^g^	0.2	0.1	-	-	-	-	-	RI, MS
66	Palustrol	1557	1922	1562	tr	tr	-	0.1	2.6	0.5	0.4	RI, MS, ^13^C NMR
67	Caryophyllene oxide	1566	1977	1568	4.1	1.4	10.8	5.5	4.6	3.6	5.1	RI, MS, ^13^C NMR
68	Spathulenol	1568	2122	1568	-	-	-	-	-	-	0.1	RI, MS
69	Humulene oxide I	1580	2009	1581 ^h^	0.1	tr	0.3	0.2	0.1	0.1	0.1	RI, MS
70	Viridiflorol *	1590	2085	1581	-	tr	0.1	-	1.4	0.4	0.2	RI, MS, ^13^C NMR
71	Humulene oxide II *	1590	2033	1597	1.6	0.1	3.6	1.6	1.0	0.5	0.7	RI, MS, ^13^C NMR
72	Eremoligenol	1614	2180	1614 ^i^	0.1	tr	0.7	0.7	-	-	-	RI, MS, ^13^C NMR
73	τ-Cadinol	1632	2164	1633 ^i^	-	-	-	0.2	tr	-	-	RI, MS
74	β-Eudesmol	1638	2222	1633	0.6	0.1	0.5	0.3	0.1	0.2	0.2	RI, MS, ^13^C NMR
75	α-Eudesmol	1652	2233	1641	-	tr	0.1	-	0.1	0.8	0.2	RI, MS, ^13^C NMR
76	β-Bisabolol	1654	2211	1659	-	-	-	-	-	0.5	-	RI, MS, ^13^C NMR
77	α-Bisabolol	1664	2210	1666 ^i^	tr	tr	-	-	0.1	0.8	0.2	RI, MS, ^13^C NMR
78	α-*epi*-Bisabolol	1666	2213	1674	-	0.2	-	-	0.3	3.7	0.3	RI, MS, ^13^C NMR
79	Benzyl benzoate	1720	2620	1733	0.1	tr	0.1	-	tr	tr	tr	RI, MS
80	*m*-Camphorene	1939	2198	1955 ^j^	-	tr	tr	tr	0.1	tr	-	RI, MS
81	*p*-Camphorene	1973	2246	1986 ^j^	-	tr	-	-	tr	-	tr	RI, MS
	Total identified				93.5	98.7	94.9	94.4	95.4	95.2	95.6	

^a^ Order of elution and relative percentages of individual components are given on apolar column (BP-1), excepted those with an asterisk (*), percentages on polar column BP-20; percentages were obtained by peak-area normalization, without using correcting factors; ^b^ Ria, RIp = retention indices measured on apolar and polar capillary columns, respectively; ^c^ RI_lit_,, literature retention indices [6] otherwise stated; ^d^ [7], ^e^ [8], ^f^ [9], ^g^ [10], ^h^ [11], ^i^ [12], ^j^ [13]; ^k^ Sse: *S. sessilifolia*; ^l^ Sle: *S. leucodermis*; ^m 13^C NMR allowed the identification of components at content as low as 0.4%; # Correct isomer not identified.

**Table 4 plants-12-01967-t004:** Data sampling.

	*Salvia sessilifolia*				*Salvia leucodermis*		
Sample	Sse1	Sse2	Sse3	Sse4	Sle1	Sle2	Sle3
Voucher	RLL 1819	ST1493	ST1514	RLL 1803	RLL 1604	RTF 162	RTF 103
Location	Tsiafakafo Ankaratra (ANK)	Andavabato, Manalalondo (ITS)	Tsiafakafo Ankaratra (ANK)	Tsiafakafo Ankaratra (ANK)	Tsiafakafo Ankaratra (ANK)	Inanobe, Faratsiho (FAR)	Aminamboara Andranomiely (ITS)
	11/2017	08/2015	09/2016	10/2016	11/2017	08/2022	09/2022
Elevation (m)	2385	1558	2203	2013	2203	2180	2300
GPS	19°20′17″ S 47°13′38″ E	19°17′18″ S 47°04′12″ E	19°18′57″ S 47°13′06″ E	19°20′49″ S 47°12′35″ E	19°18′57″ S 47°13′06″ E	19°31′03″ S 47°04′20″ E	19°21′25″ S 47°14′01″ E

## Data Availability

The data presented in this study are available on request from the corresponding author.

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
