# Peer review of "Extinction Risk Assessment and Chemical Composition of Aerial Parts Essential Oils from Two Endangered Endemic Malagasy Salvia Species"

_plants, 2023, doi:10.3390/plants12101967_

Round 1

Reviewer 1 Report

The manuscript  Extinction Risk Assessment and Chemical Composition of Aerial Parts Essential Oils from Two Endangered Endemic Malagasy Salvia Species is very interesting and very well-written article.

Please write the name of plant name in italic "Salvia" in the title

please changes the sentence 'the Lamiaceae contains 236 genera belonging to 7 173 species" to

the Lamiaceae contains 7 173 species belonging to 236 genera

More information is needed about the amount of plant material subjected to hydrodistillation and the yield of the essential oil at least of one sample of each species.

What is the justification of the use of NMR for some well-known components of the essential oil, which have very high percentage of matching with library charts.

As stated in the article title I didn't find a relationship between extinction risk assessment and the composition of the essential oils, however the authors should propose a strategy of saving the endangered plants by the study of their essential oil.  

please inset at least one chromatogram chart of each Salvia species in the supplementary material.

Author Response

Dear Editor,

We wish to submit a revision of the manuscript. We considered all the suggestions and requirements of reviewers. We revised the manuscript accordingly with all changes listed below.

We hope that this manuscript is well suited for publication in Plants.

Referee 1

The manuscript  Extinction Risk Assessment and Chemical Composition of Aerial Parts Essential Oils from Two Endangered Endemic Malagasy Salvia Species is very interesting and very well-written article.

Please write the name of plant name in italic "Salvia" in the title

Done

please changes the sentence 'the Lamiaceae contains 236 genera belonging to 7 173 species" to the Lamiaceae contains 7 173 species belonging to 236 genera.

Done

More information is needed about the amount of plant material subjected to hydrodistillation and the yield of the essential oil at least of one sample of each species.

Two sentences with several information are added in paragraph « 3.3 3 Essential Oil Isolation

What is the justification of the use of NMR for some well-known components of the essential oil, which have very high percentage of matching with library charts.

We agree with the referee. This essential oil is not very complex. However, various sesquiterpenes, usually found in these essential oils, exhibit insufficiently differentiated mass spectra, as could be seen with the spectra of a-bisabolol and epi-a-bisabolol. Moreover, these two compounds have the same RI on non-polar column and a very close RI to polar column and therefore their identification by GC-MS in combination with RIs remains a difficult task. In this case, 13C NMR analysis is a usefulness technique to identify epimers.

373

As stated in the article title I didn't find a relationship between extinction risk assessment and the composition of the essential oils, however the authors should propose a strategy of saving the endangered plants by the study of their essential oil.

Sorry if our text has been misleading. Our study focused on an endangered species, so we have studied its “population status” in relation to its ecology. In addition, we took the opportunity to study also the chemical composition of its EO with limited impact on the populations.

In the situation that an original/interesting EO composition was found, it would have been a supplementary argument for its preservation.

please inset at least one chromatogram chart of each Salvia species in the supplementary material.

A chromatogram is added.

Referee 2

The proposed manuscript “Extinction Risk Assessment and Chemical Composition of Aerial Parts Essential Oils from Two Endangered Endemic Malagasy Salvia Species” is based on an idea with scientific and practical importance, the experiment is relatively well designed and performed, the manuscript is well composed and written, and the results are clearly presented. Therefore I could suggest minor revisions to be done before accepting for publication, namely:

In Table 3, only experimentally determined RI values (on BP-1 and BP-20 columns) are listed, for avoiding misidentification, the literature data should be included, please add. 

A column with literature data  on non-polar column is added.

In addition, I would recommend the representative chromatograms of the studied samples to be included as a Figure.

A chromatogram is added.

Referee 3

The manuscript entitled "Extinction Risk Assessment and Chemical Composition of Aerial Parts Essential Oils from Two Endangered Endemic Malagasy Salvia Species" presents results dealing with the determination of the chemical composition of two sage species. The following points should be taken into account:

* The originality of the work should be clearly indicated in the introduction section

 A sentence has been added at the end of the introduction.

* The botanical identification of the used species should be indicated

 Some adequate information are added in paragraph “3.1 Plant material”.

* More details on the identification of the indicated compounds are needed

For the identification we have followed the same methods that we used for the last 20 years (references 24-26). These methods are classical in analytical chemistry: combination of retention indices, mass spectra and NMR chemical shifts as fully detailed in the paragraph “3.6. Identification of Individual Components”.

* Chromatographic profiles of the obtained essential oils will be useful

A chromatogram is added.

Reviewer 2 Report

The proposed manuscript “Extinction Risk Assessment and Chemical Composition of Aerial Parts Essential Oils from Two Endangered Endemic Malagasy Salvia Species” is based on an idea with scientific and practical importance, the experiment is relatively well designed and performed, the manuscript is well composed and written, and the results are clearly presented. Therefore I could suggest minor revisions to be done before accepting for publication, namely:

            In Table 3, only experimentally determined RI values (on BP-1 and BP-20 columns) are listed, for avoiding misidentification, the literature data should be included, please add.  

            In addition, I would recommend the representative chromatograms of the studied samples to be included as a Figure.

Author Response

(The authors gave the same response as above.)

Reviewer 3 Report

The manuscript entitled "Extinction Risk Assessment and Chemical Composition of Aerial Parts Essential Oils from Two Endangered Endemic Malagasy Salvia Species" presents results dealing with the determination of the chemical composition of two sage species. The following points should be taken into account:

* The originality of the work should be clearly indicated in the introduction section

* The botanical identification of the used species should be indicated

* More details on the identification of the indicated compounds are needed

* Chromatographic profiles of the obtained essential oils will be useful

Author Response

(The authors gave the same response as above.)

Round 2

Reviewer 3 Report

Could be accepted in the current form